# Sodium Intake and Target Organ Damage in Hypertension—An Update about the Role of a Real Villain

**DOI:** 10.3390/ijerph17082811

**Published:** 2020-04-19

**Authors:** Federica Nista, Federico Gatto, Manuela Albertelli, Natale Musso

**Affiliations:** Unit of Hypertension, Clinical Endocrinology, Department of Internal Medicine, Ospedale Policlinico San Martino Genova, University of Genoa Medical School, 6-16132 Genoa, Italy; nistafodi@libero.it (F.N.); fedgatto@hotmail.it (F.G.); manuela.albertelli@unige.it (M.A.)

**Keywords:** sodium intake, organ damage, hypertension, cardiovascular risk

## Abstract

Salt intake is too high for safety nowadays. The main active ion in salt is sodium. The vast majority of scientific evidence points out the importance of sodium restriction for decreasing cardiovascular risk. International Guidelines recommend a large reduction in sodium consumption to help reduce blood pressure, organ damage, and cardiovascular risk. Regulatory authorities across the globe suggest a general restriction of sodium intake to prevent cardiovascular diseases. In spite of this seemingly unanimous consensus, some researchers claim to have evidence of the unhealthy effects of a reduction of sodium intake, and have data to support their claims. Evidence is against dissenting scientists, because prospective, observational, and basic research studies indicate that sodium is the real villain: actual sodium consumption around the globe is far higher than the safe range. Sodium intake is directly related to increased blood pressure, and independently to the enlargement of cardiac mass, with a possible independent role in inducing left ventricular hypertrophy. This may represent the basis of myocardial ischemia, congestive heart failure, and cardiac mortality. Although debated, a high sodium intake may induce initial renal damage and progression in both hypertensive and normotensive subjects. Conversely, there is general agreement about the adverse role of sodium in cerebrovascular disease. These factors point to the possible main role of sodium intake in target organ damage and cardiovascular events including mortality. This review will endeavor to outline the existing evidence.

## 1. Introduction

Table salt is mainly sodium chloride. The active ion in cardiovascular pathophysiology is sodium. Salt and sodium are in a 2.54:1 w/w ratio, the molecular weight of sodium is 23, so 100 mEq/mmol of sodium have a weight of 2300 mg or (2.3 × 2.54) 5.84 g of salt. Sodium consumption is directly related to blood pressure. The restriction of sodium intake is reported as a mandatory non-pharmacological measure to decrease blood pressure in the international Guidelines [1,2].

The vast majority of papers in the literature show the advantages of a low sodium approach for treating hypertension [3,4,5,6,7], while a minority manages to cast doubts on the healthy effects of reduced sodium intake [8]. Dissenting scientists have been accused of designing their studies with inherent biases in their protocols [9] in order to cloud the advantages of a general lowering of dietary sodium [10].

While the favorable effects of salt (and thus sodium) restriction in the diet are substantially accepted by the scientific world, less is known about the specific effects of sodium consumption (and the possible advantages of dietary sodium lessening) on target organ damage and cardiovascular events [11].

In spite of the pioneering studies by Walter Kempner [12] in the middle of the last century, where exceptional effects were attained in terms of cardiac mass reduction with a low sodium approach (a very low one, indeed), we had to wait until the 1980s to read about the first echocardiographic reports in this field [13].

Since then, research about dietary regression of left ventricular hypertrophy (LVH) has been heavily outweighed by studies based on pharmacological approaches [14].

Similarly, the favorable effects of sodium intake restriction in related fields such as stroke prevention, proteinuria reduction, and cardiovascular prognosis improvement do not represent a large mass of published papers [11].

This review will endeavor to evaluate the healthy effects of a decrease in dietary sodium on the hard endpoints already shown in the present literature.

## 2. Methods

Papers were retrieved from PubMed (NCBI), ISI Web of Knowledge^®^/Web of Science/Science Citation Index (Clarivate), and Cochrane^©^ databases (Wiley) by means of the usual search strings and simple Boolean operators for blood (arterial) pressure, hypertension, salt (sodium), diet (intake, consumption), left ventricular hypertrophy (mass), stroke, proteinuria, cardiovascular prevention, and cardiovascular events (coronary heart disease, unstable angina, revascularization procedures, congestive heart failure, and stroke).

The searches retrieved 968 citations when duplicates were removed (Figure 1). Moreover, 752 papers were excluded after reading their titles and abstracts, and because they were outside the scope of this study. One hundred and seventy-seven full papers were reviewed. Of these, 132 papers in journals with a certified impact factor, with at least an English abstract, and published from 1980 onwards (except for two of the earlier papers that focused on the aim of our study) were selected. Papers that had already been discussed in cited reviews were not included, and instead reviews were included. In addition to the retrieved references were the Guidelines [1,2].

## 3. Discussion

### 3.1. Sodium Intake and Left Ventricular Hypertrophy

LVH is generally recognized as a primary marker of high cardiovascular risk and a potent predictor of mortality, in both normotensive and hypertensive patients [2]. LVH regression [16], or reduction [17], was shown to occur after blood pressure (BP) control as early as the beginning of the echocardiographic era [18], but the decrease in left ventricular mass (LVM) seemed already not to be entirely dependent on BP reduction [16,18]. The regression of LVH towards a normal LVM is considered to be a goal of antihypertensive treatment, because the LVH regression itself represents a marked reduction and a strong improvement in the cardiovascular risk [2].

The diet plays a key role in the reduction of the LVM. Already in the middle of the last century, a (very) low sodium approach could sensibly reduce cardiac mass in patients with malignant hypertension [12]. Thereafter, and more recently, the dietary approach, based on sodium intake reduction, showed positive results in terms of LVM decrease [3,19,20,21,22,23,24,25,26,27]. While sodium restriction has favorable effects, a high sodium diet seems to be a strong and independent predictor of LVH [24,28,29,30,31,32,33,34,35,36]. Similarly, while a successful lowering of BP may generally appear more important than the type of approach in inducing the reversal of LVH [37], an LVM reduction after diet and/or drugs seems unrelated to BP values in the view of other researchers [13]. Moreover, the regression of LVH with active drug treatment with the inhibition of the renin–angiotensin system is impaired by high sodium intake [22]. In a relatively large data set the BP-independent effect of a low sodium diet on the LVM has been observed, and a substantial LVH reversal has been shown [38]. Even in a secondary hypertension setting, the role of sodium intake plays a pivotal role in the enlargement of cardiac dimensions [39]. In spite of the clear dietary sodium involvement, it is less clear whether or not sodium intake may be linked to the LVM in terms of sodium sensitivity or sodium resistance [40]. Incidentally, sodium intake appears to be related to arrhythmias [41], or at least to an increased susceptibility to arrhythmias [42].

The results of what we have learned from the literature point to a key role for sodium intake in promoting both an increase in BP and (possibly with an independent role) an enlargement of LVM, with or without the involvement of traditional growth factors, such as the mediators of the sympathetic or the renin–angiotensin–aldosterone system, on cardiac myocytes [43,44].

The response of BP to dietary sodium shows a normal distribution across populations. Many (so-called salt-sensitive) individuals (represented mainly by women, older individuals, and black people), take advantage of sodium restriction while others (salt-resistant, i.e., men, young people, and white people) do not [45]. Pre-hypertensive and hypertensive subjects usually benefit most [45].

### 3.2. Sodium Intake and Cardiovascular Events

Sodium intake is linked to an increase in BP [5], to the incidence of hypertension [6], is associated with endothelial damage [46], metabolic disturbances [7,46], and markers of subclinical coronary disease [47,48]. Furthermore, a decrease in sodium intake has favorable effects on both the BP itself [49] and arterial stiffness [50].

Mechanisms linking sodium intake and BP are represented mainly by volume expansion and interactions with the renin–angiotensin–aldosterone system (RAAS), even if they are not completely understood [51]. Other research is in progress exploring the sodium–immune system connections [52,53,54]. Sodium and the immune system can impact BP through the modulation of myeloid cells as well as lymphocytes [53]. Moreover, alimentary sodium promotes endothelial dysfunction, and cytokine secretion [54]. Macrophages and endothelial growth factors may be involved as well [55]. All these events play a key role in the pathophysiology of initial cardiovascular damage [54].

Effects other than hypertension have been attributed to an exaggerated sodium intake. These include direct damage to arteries through endothelial dysfunction and reduced nitric oxide bioavailability, or by increasing vasoconstrictor peptides [55]. Other direct effects (not mediated by an increased BP) include the involvement of epithelial sodium channels in the distal nephrons [55].

These mechanisms provide some pathophysiological bases for the increased risk of stroke [56,57], stroke mortality [58], coronary heart disease mortality [57], and overall cardiovascular risk [57] associated with increased sodium consumption. Even in experimental models, a high sodium diet produces left ventricular injury and probably causes myocardial ischemia [59].

Consequently, it is not surprising to find mounting evidence linking sodium intake to a greatly increased cardiovascular and cerebrovascular risk [46,60,61,62,63].

Recently, a Dutch group expressed a word of caution, when they observed an increased risk of stroke in association with (very) low excretion rates of sodium in a large cohort of patients [64], even if many confounders had not been taken into account [65].

Finally, it is worth noting that excess sodium intake increases the overall cardiovascular mortality [66], at least since publication of the NutriCode results. This large meta-analysis of more than 100 trials showed the devastating effects of excess sodium in the diet: from 1.1 to 2.2 million annual deaths worldwide, in 2010, were attributable to dietary sodium above the WHO reference level of 2.0 g a day; this figure represents 9.5% of all cardiovascular deaths [66,67]. The majority of these deaths were due to coronary heart disease (41.7%) and to stroke (41.6%) [67].

A cornerstone of cardiovascular morbidity and mortality prevention is the implementation of sodium restriction by clinicians [68,69], both in high-risk patients and in healthy individuals [70]. Cardiovascular disease is linked to sodium intake [71,72,73], and restricting sodium consumption has favorable effects [74,75,76], while studies flawed by incorrect urinary sodium sampling show mixed or non-significant results [77,78,79], again returning to the J-shaped relationship [79].

Even in congestive heart failure, the importance of the dietary approach based on a low sodium intake should be remembered [80].

The relationships between sodium intake and BP represent the basis for the preventive effects of sodium restriction on cardiovascular morbidity and mortality [81]. Nevertheless, some doubts have been expressed about the real effects of low sodium intake on cardiovascular morbidity [82]. A further review examined 17 studies comprising more than 3000 patients evaluated for nutritional interventions in congestive heart failure (CHF), which represents an ever-increasing cardiovascular event as well as a source of cardiovascular morbidity and mortality. Although a low sodium diet may improve morbidity in this clinical setting, this research showed that a low sodium intake might be harmful [83].

Despite this evidence, many others recommend sodium restriction in CHF patients, recognizing the effects of dietary interventions on the heart and blood vessels [84,85,86], although this is not generally appreciated.

During the last few years, some reviews have addressed the pathophysiological problems connecting alimentary sodium and cardiac damage. High importance has been credited to aldosterone, parathyroid hormone, and to interstitial (tissue) sodium [87,88,89]. Finally, and more recently, many studies were scrutinized to assess the benefits of a dietary approach in CHF [83,90,91,92,93,94] with mixed results. The conclusions in this field remain unclear, and further research is required to verify the importance of dietary strategies on heart failure [91].

Cerebrovascular diseases seem more dependent on sodium intake. Current Guidelines recommend a low sodium approach to prevent cardiovascular morbidity and mortality. There is robust evidence to show that a diet that includes sodium restriction can prevent stroke [95]. Health professionals should take an active role in promoting a low sodium diet to reduce the stroke burden [96].

A large number of reviews and meta-analyses support a low sodium approach for the prevention of stroke and conversely show the relationship between high sodium intake and the prevalence of stroke. The advantages of a low sodium diet are evident in both normotensive and hypertensive subjects, with more evidence in the latter [5,97,98,99,100,101,102,103]. Similarly, notable evidence supports the link between high sodium intake and the occurrence of cerebrovascular disease [63,104,105,106,107,108,109].

The mechanisms that connect alimentary sodium to stroke have been discussed in recent years and range from vascular lesions with enhanced neurogenesis [110], to endothelial dysfunction [111], inflammation due to low parasympathetic activity [112], small vessel disease associated with high sodium consumption [113], adenosine receptor expression [114] or other protein expression, and microglia polarization [115].

### 3.3. Sodium Intake and Renal Impairment/Proteinuria

Updated Guidelines suggest a low sodium approach for the prevention of chronic kidney disease (CKD) or its progression. Nevertheless, even in this field discussions do exist. A recent large quantitative review [116] found that robust evidence is lacking on the association between sodium intake and CKD delay or prevention.

Other researchers [117] found high-quality studies showing adverse effects of alimentary sodium on health outcomes such as CKD.

To add fuel to the discussion, a recent meta-analysis of 11 trials involving hundreds of CKD patients [118] showed how sodium restriction induces BP reduction and proteinuria improvement in CKD patients.

Others showed how high sodium (and low water) intake have a pathogenic role in hypertension and CKD [119], whereas a few years ago [120] the importance of a low sodium approach to increase the protective effects of RAAS blockade in CKD patients was shown.

So, even with some warning, nephrologists are advised to support their patients with carefully tailored diets that are low in sodium [121]. Support for this view comes from a large prospective community-based study involving more than 8000 patients [122]. Here the authors showed a U-shaped relationship between sodium intake and the risk of CKD development, with maximum risk either in patients with very low (under 2.08 g/day) or high sodium intake (more than 4.03 g/day).

A large Cochrane review [123] concluded that sodium restriction can reduce both BP and proteinuria in CKD patients, even if long-term studies showing the direct effects of a low sodium approach on the progression to end-stage kidney disease are not available.

Short-term studies again showed that sodium reduction can help the antiproteinuric effect of RAAS blockade, but in the vast majority of CKD and renal transplant patients sodium consumption is close to that in the general population and remains high [124].

More recently, multiple dietary approaches, including sodium restriction, have been associated with lower mortality in CKD patients [125].

Dietary sodium, besides influencing fluid volumes, can induce tissue remodeling and activate the immune system [126]; this in turn may be linked to the inflammatory and fibrotic effects that have a role in the pathophysiology of kidney disease.

In this latter field, while traditional biomarkers such as creatinine appear to be decreasing in importance, new biomarkers are gaining ground in experimental studies, capable of linking sodium intake and early kidney damage [127].

## 4. Conclusions

A high sodium intake increases BP, overall cardiovascular adverse outcomes, and cardiovascular mortality, as well as the risk of stroke. Excess sodium induces worsening of urinary albumin excretion and enlargement of cardiac mass.

As already observed in the middle of the last century, dietary sodium can decrease survival in experimental animals [128]. Excess sodium is responsible for one cardiovascular-related death out of 10 worldwide [67]. Conversely, sodium restriction can reduce the rates of cardiovascular events by at least 25% [129].

Sodium restriction, as suggested in the literature and Guidelines, appears to be a safe measure, which significantly decreases BP [123,130], antihypertensive drug consumption [123,130], and the global cardiovascular risk [131]. Stroke prevention should be based on a dietary approach involving a decrease in dietary sodium [132]. The European Salt Action Network supported sodium reduction on a population basis, to improve the risk of cardiovascular diseases [133]. This is in agreement with the positions of scientific societies, Guidelines and WHO. Population-wide policies are now actively implementing sodium restriction measures around the globe, with good results being achieved in many cases [134], involving both upstream and downstream interventions.

Even a moderate decrease in sodium consumption can result in major successes in terms of adverse event prevention and pharmacoeconomics [135].

## Figures and Tables

**Figure 1 ijerph-17-02811-f001:**
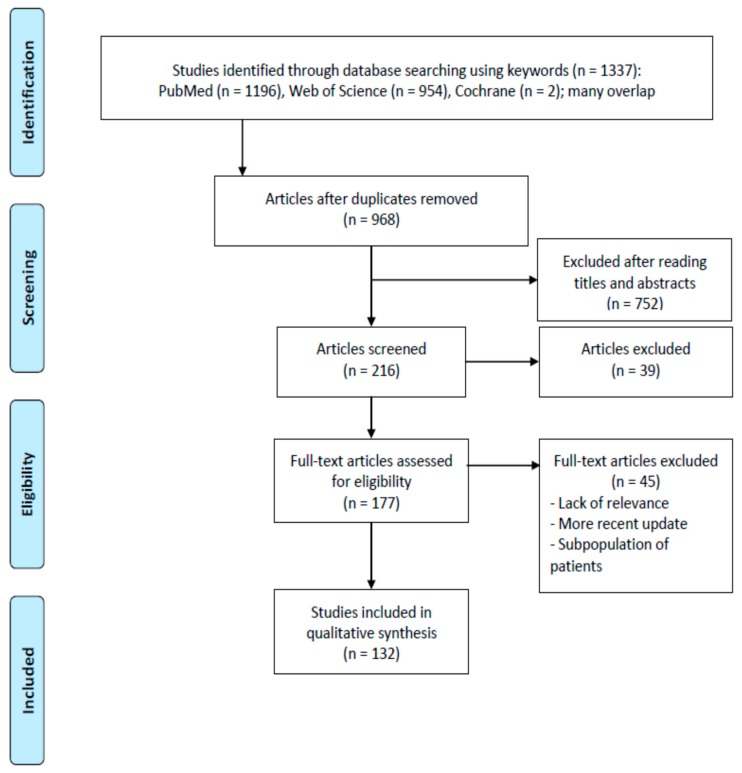
PRISMA 2009 flow diagram illustrating the selection process for the articles included in this review [15]. PRISMA documents are distributed under the terms of the Creative Commons Attribution License, which permits unrestricted use, distribution and reproduction.

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
