# Peer review of "Sodium Intake and Target Organ Damage in Hypertension—An Update about the Role of a Real Villain"

_ijerph, 2020, doi:10.3390/ijerph17082811_

Round 1

Reviewer 1 Report

The abstract should bring more summarized information on the target organ damage associated to hypertension, which is the primary goal of the manuscript. The strong wording of certain sentences should be revised, such as in “In spite of this seemingly unanimous consent, some researchers manage to inflate their view, about the unhealthy effects of a reduction of salt intake, with insidious data.”  

As a general comment, the use of terms as salt and sodium should be standardized over the whole manuscript.

In the discussion, subsection 3.1, on Left Ventricular Hypertrophy is well developed, nevertheless the other subsections need improvements:

  • In 3.2, part of the arguments should be moved to the conclusions, such as the CVD prevention policies and the strategies for sodium reduction. In that sense, there is no discussion on public policies for salt reduction, which are the most effective strategies for population-wide reduction. The first paragraphs of this subsection (Lines 100-107), on the physiologic mechanisms of sodium intake, hypertension, and cardiovascular disease
  • In 3.3 (lines 130-139), the relationship between salt intake and renal impairment/proteinuria is not sufficiently developed, compared to the previous subsections of the manuscript.

The conclusions need to better summarize the information on the discussion section and to analyze the gaps in research, the strategies for sodium reduction and the strengths and weaknesses of the analysis.

Author Response

Dear Sir, we really appreciated your comments that we feel offered in a constructive spirit.

Changes from the previous version are marked in red. References have been added accordingly.

We have revised our abstract with a view to including clear information as for the main purpose of the review. For this reason, we added a short clarification (rows 23-28) on the role of sodium intake directly related to increased blood pressure and consequently to target organ damage.

Moreover, as suggested, the sentences or words not in line with the appropriate wording used in the whole review, were removed (rows 18-19 and row 41).

In the introduction, we have pointed out the scientific meaning of the terms salt and sodium (rows 34-36). The word sodium has been substituted for salt throughout the review.

In the first paragraphs of section 3.2, we added a short and concise description of the pathophysiological mechanisms which connect sodium intake with target organ damage (rows 118-129). Furthermore, in the same section, we managed to improve the discussion as for congestive heart failure and cerebrovascular disease (rows 153-185).

We revised section 3.3 in order to point out the current evidences as for salt intake and renal impairment (rows 198-229).

The conclusions were enlarged to highlight the importance of the strategies adopted for sodium reduction worldwide(rows 241-246).

The differences and discrepancies being present among the articles analyzed were discussed in each revised subsection of the review.

Reviewer 2 Report

Authors described detailed analyses about implmentation of salt intake for target organ damage by extensive reviews for literatures.Their comments for left ventricular hypertrophy, cardiovascular events, and renal involvements are acceptable. Reviewer ask the authors to provide additional mathematical conjunctions of literatures, because their analyes remain not quantitatively but qualitatively. If they provide mor information about their survey with substantial data, this manuscript is getting more better for readers of the journal.

Author Response

Dear Sir,

The main scope of this review is to describe qualitatively the role of sodium intake as for organ damage in hypertension. In this way, we did not focus our synthesis on the quantitative analysis of the literature. Moreover, we have considered that providing a quantitative review might change our beginning project and it seems difficult to be accomplished in the short time span allowed by the Editors to revise our paper.

Anyway, we managed to improve the readability and the accuracy of our paper as much as possible.

Changes from the previous version are marked in red. References have been added accordingly.

Reviewer 3 Report

In the present study, authors have focused on the role of salt intake on the development of hypertension and hypertension-related complications. The authors presented a short review of the most important clinical findings indicating a strong correlation between increased salt consumption and hypertension development. However, in my opinion, authors should also describe shortly the proposed mechanisms through which an increased sodium intake causes hypertension, including its role as a factor impairing the RAA system, but not only. There is a growing body of evidence that salt intake may activate the immune system, which contributes to an increase in blood pressure. The authors should also mention the different responses to an increased salt intake among the population and, in consequence, the occurrence of salt-sensitive and salt-resistant responders, respectively.

Author Response

Dear Sir,

we really appreciated your comments that we feel offered in a constructive spirit.

Changes from the previous version are marked in red. References have been added accordingly.

In subsection 3.2 we added a brief explanation to underline the complex and not completely understood mechanisms connecting sodium intake and hypertension. Furthermore, we focused on the thesis considering the deep link among sodium intake, hypertension and immune system (rows 118-129).

The different responses between salt-sensitive and salt-resistant have been mentioned at the end of 3.1 section (rows 108-112), with what seems an appropriate reference.

Round 2

Reviewer 1 Report

The issue of adverse cardiovascular outcomes of sodium intake is a global concern and this work summarizes and updates the impact of excessive sodium on target organs. The manuscript has improved considerably and has incorporated all suggestions. 

Reviewer 2 Report

As stated by the author, limitation of the paper is clear. However they performed systematic review for salt intake and organ damages, lack of statitical data seemed critical shortness for the paper. This should be considered limited significance for this field. It might be of interest with such a limitation for the readers of the journal.